# Chronic Lung Allograft Dysfunction Is Associated with Increased Levels of Cell-Free Mitochondrial DNA in Bronchoalveolar Lavage Fluid of Lung Transplant Recipients

**DOI:** 10.3390/jcm11144142

**Published:** 2022-07-16

**Authors:** Emmanuel Schneck, Ingolf Askevold, Ramona Rath, Andreas Hecker, Martin Reichert, Stefan Guth, Christian Koch, Michael Sander, Werner Seeger, Konstantin Mayer, Winfried Padberg, Natascha Sommer, Stefan Kuhnert, Matthias Hecker

**Affiliations:** 1Department of Anesthesiology, Operative Intensive Care Medicine and Pain Therapy, Justus Liebig University of Giessen, 35392 Giessen, Germany; emmanuel.schneck@chiru.med.uni-giessen.de (E.S.); christian.koch@chiru.med.uni-giessen.de (C.K.); michael.sander@chiru.med.uni-giessen.de (M.S.); 2Department of General and Thoracic Surgery, University Hospital Giessen, Justus Liebig University of Giessen, 35392 Giessen, Germany; ingolf.askevold@chiru.med.uni-giessen.de (I.A.); andreas.hecker@chiru.med.uni-giessen.de (A.H.); martin.reichert@chiru.med.uni-giessen.de (M.R.); winfried.padberg@chiru.med.uni-giessen.de (W.P.); 3Department of Internal Medicine II, Universities of Giessen and Marburg Lung Center (UGMLC), Member of the German Center for Lung Research (DZL), Justus-Liebig University of Giessen, 35392 Giessen, Germany; ramona.rath@med.uni-giessen.de (R.R.); werner.seeger@innere.med.uni-giessen.de (W.S.); konstantin.mayer@innere.med.uni-giessen.de (K.M.); natascha.sommer@innere.med.uni-giessen.de (N.S.); stefan.kuhnert@innere.med.uni-giessen.de (S.K.); 4Department of Thoracic Surgery, Kerckhoff Heart and Lung Center, 61231 Bad Nauheim, Germany; s.guth@kerckhoff-klinik.de; 5Excellence Cluster Cardio-Pulmonary Institute (CPI), 35392 Giessen, Germany; 6Institute for Lung Health (ILH), 35392 Giessen, Germany; 7Department of Pulmonary and Sleep Medicine, ViDia Hospitals, 76137 Karlsruhe, Germany; 8Department of Internal Medicine, Pulmonary and Critical Care Medicine, University Hospital of Giessen and Marburg (UKGM), Klinikstr. 33, 35392 Giessen, Germany

**Keywords:** transplantation, inflammation, CLAD, BOS, mitochondria

## Abstract

Chronic Lung Allograft Dysfunction (CLAD) is a life-threatening complication that limits the long-term survival of lung transplantation patients. Early diagnosis remains the basis of efficient management of CLAD, making the need for distinctive biomarkers critical. This explorative study aimed to investigate the predictive power of mitochondrial DNA (mtDNA) derived from bronchoalveolar lavages (BAL) to detect CLAD. The study included 106 lung transplant recipients and analyzed 286 BAL samples for cell count, cell differentiation, and inflammatory and mitochondrial biomarkers, including mtDNA. A receiver operating curve analysis of mtDNA levels was used to assess its ability to detect CLAD. The results revealed a discriminatory pro-inflammatory cytokine profile in the BAL fluid of CLAD patients. The concentration of mtDNA increased in step with each CLAD stage, reaching its highest concentration in stage 4, and correlated significantly with decreasing FEV1. The receiver operating curve analysis of mtDNA in BAL revealed a moderate prediction of CLAD when all stages were grouped together (AUROC 0.75, *p*-value < 0.0001). This study has found the concentration mtDNA in BAL to be a potential predictor for the early detection of CLAD and the differentiation of different CLAD stages, independent of the underlying pathology.

## 1. Introduction

Lung transplantation is the last therapeutic option for patients suffering from end-stage pulmonary diseases such as Idiopathic Pulmonary Fibrosis (IPF) or Chronic Obstructive Pulmonary Disease (COPD). Over the last decade, advances in transplantation surgery, intensive care medicine, and immunosuppressive therapy have resulted in a significant increase in the survival rates of adults after primary lung transplantation. Despite this progress, long-term survival after lung transplantation is still significantly reduced compared to other organ transplantations, with a mean five-year survival rate between 50 and 60% [1,2,3,4]. In addition to early complications such as surgical site infections, pneumonia, and acute allograft rejection, the development of Chronic Lung Allograft Dysfunction (CLAD) is a significant limiter for the long-term survival of patients after lung transplantation. 

CLAD is defined as a *“substantial and persistent decline (≥ 20%) in measured FEV1 value from the reference (baseline) value*” [5] and can be divided into two primary pathophysiological etiologies: since the symptoms of CLAD, such as dyspnea, reduced respiratory capacity, and bronchial obstruction, resemble those of bronchiolitis, CLAD was first described as a Bronchiolitis Obliterans Syndrome (BOS). However, several additional CLAD etiologies have subsequently been described, including restrictive pulmonary disorders that led to the introduction of Restrictive Allograft Syndrome (RAS) in 2011 [6]. Associated with a five-year prevalence of 50% after lung transplantation, both phenotypes of CLAD are clinically relevant, with BOS presenting more frequently than RAS [4]. However, RAS is more lethal compared to BOS [6]. Furthermore, survival rates after CLAD diagnosis remain poor due to a paucity of effective treatment options. Possible therapeutic approaches include a modification of the patient’s immunosuppressive regime, treatment with azithromycin or montelukast, or the initiation of extracorporeal photophoresis (ECP) [7,8,9,10,11]. However, prophylaxis for infections, detection of acute allograft rejections with subsequent treatment, and early diagnosis form the basis of efficient CLAD management.

While spirometry, computed tomography scans, and lung biopsies enable a specific CLAD diagnosis and remain the current gold diagnostic standard, they do not allow for a diagnosis before the onset of clinical symptoms. Several biomarkers have been evaluated for primary graft rejection, but they lack sufficient predictive power to identify CLAD [12]. However, liquid biopsies derived from blood or bronchoalveolar lavages (BAL) may provide a solution due to their minimal invasiveness and improved specificity and sensitivity. In transplantation medicine, liquid biopsies consist primarily of specific DNA fragments from the donor and the transplanted organ [13]. A recent prospective study of lung transplanted patients identified the plasma ratio of donor-derived cell-free DNA to total cell-free DNA as a potential biomarker for detecting primary organ rejection [14]. Moreover, an evaluation of BAL-derived liquid biopsies efficiently predicted CLAD onset and mortality [15].

Although mitochondrial DNA (mtDNA) has been evaluated as a biomarker for cancerous and infectious diseases, its potential as a biomarker for identifying CLAD has yet to be evaluated [16,17]. Moreover, mtDNA might be of interest in CLAD because it plays an important role in acute organ rejection following lung transplantation, and elevated levels of mtDNA are associated with primary graft rejection. One potential explanation for this association is formylated peptide receptor-mediated neutrophil trafficking causing reactive oxygen species (ROS)-induced pulmonary edema [18]. Since ROS activation is also closely connected to CLAD pathology, the investigation of mtDNA as a predictive biomarker for CLAD is merited, particularly donor-derived mtDNA. However, the association between BAL-derived mtDNA encoding for NADH dehydrogenase 1 (ND1) and the onset of CLAD and its severity has not yet been described. Here, we evaluate the correlation between ND1-mtDNA levels and CLAD and the potential association between elevated ND1-mtDNA and higher morbidity.

## 2. Methods

### 2.1. Study Design

This single-center retrospective data analysis was approved by the local ethical committee of the Justus-Liebig-University of Giessen (Protocol number 187/14). Patient consent was obtained during preoperative enrollment for lung transplantation within the Pulmonary Department at the University Hospital of Giessen. Transplantations were performed at the University Hospital of Giessen and the Department of Thoracic Surgery at the Kerckhoff Clinic in Bad Nauheim. Only patients who underwent lung transplantation due to end-stage pulmonary disease between January 2010 and June 2019 and received bronchoscopy with BAL between September 2015 and June 2019 were included in the study.

### 2.2. Management of Lung Transplantation and Surveillance

Lung transplantation was performed for all patients using standard techniques and current protocols for organ harvest and preservation, as previously described [3,19]. The preoperative assessment of the donor organ quality included bronchoscopic and macroscopic evaluation and functional oxygenation capacity under 100% oxygen insufflation. A post-operative triple-drug immunosuppressant regimen consisted of tacrolimus (trough concentration, 8–10 µg/L), mycophenolate mofetil (2 × 1500 mg/d), and prednisolone. Based on the spirometry results, CLAD was categorized according to the pulmonary council of the International Society of Heart and Lung Transplantation (ISHLT) into five stages: stage 0: FEV_1_ > 80%; stage 1: FEV_1_ 66–80% baseline; stage 2: FEV_1_ 51–65%; stage 3: FEV_1_ 35–50%; and stage 4: FEV_1_ < 35% (referred to baseline best FEV_1_) [5].

For early detection of complications, especially acute allograft rejection, surveillance bronchoscopies with bronchoalveolar lavage (BAL) and transbronchial lung biopsies were performed on the 1st, 3rd-, 6th-, 9th-, and 12th-month post-transplantation and yearly thereafter. BAL performance was evaluated with 150 mL of sterile saline in eight aliquots. Liquid recovery after instillation was performed manually with a 20 mL syringe. All recovered fluid was pooled, and aliquots were subjected to total and differential cell count analyses, microbiology, and pathology. An aliquot of 5–10 mL of BAL fluid was submitted to our biobank for further experimental analysis in accordance with the patient´s consent.

### 2.3. Quantification of mtDNA

After thawing, 200 µL of each sample was diluted in a ratio of 1:1 with phosphate-buffered saline (PBS) and briefly vortexed before 300 µL of the supernatant was collected and centrifugated for 15 min at 18,000× *g*. Next, DNA was extracted and purified using the QIAamp DNA Blood Mini Kit according to the manufacturer’s instructions (Qiagen, Venlo, The Netherlands). Prior to qPCR analysis, each purified DNA sample was diluted at 1:20 with nuclease-free, deionized–distilled water. The level of mtDNA in each sample was quantified using the following primers for NADH dehydrogenase (ND1) that were synthesized by Eurofins (Luxembourg, Luxembourg): 5′-CCA CCT CTA GCC TAG CCG TTT A-3′(ND1 mtDNA FW) and 5′-GGG TCA TGA TGG CAG GAG TAA T-3′ (ND1 mtDNA RW). The ND1-mtDNA level of each sample was measured in triplicate, and the average level was converted to copies/μL as previously described [20] based on a standard curve constructed with serial dilutions of a plasmid containing human ND1 mtDNA (OriGene Technologies, Rockville, MD, USA). The number of plasmid copies in the source was first determined with a NanoDrop 2000 spectrophotometer (Thermo Fisher Scientific, Waltham, MA, USA), and dilutions containing 30–300,000 copies of the plasmid were measured. All qPCR analyses were performed using the StepOnePlus Real-Time PCR System (Thermo Fisher Scientific, Waltham, MA, USA).

### 2.4. Clinical Data Recruitment and BAL Analyses

Clinical data were recorded during the transplantation process and included in the study. In addition to evaluating the patient´s history of pre-existing diseases and smoking, the results of BAL, spirometry, laboratory, and blood gas analyses were obtained from the hospital patient data management system. The concentrations of interleukin 6 (IL6) and 8 (IL8) and MIP 1α were quantified using an Enzyme-Linked Immunosorbent Assay (ELISA; R&D Systems, Wiesbaden, Germany) according to the manufacturer´s instructions. Note that we were unable to obtain measurements for each protein in all samples.

### 2.5. Statistical Analysis

The distribution of the data set was assessed with the Shapiro–Wilk test. Normally distributed data are reported as mean ± standard deviation or percentage, and non-normally distributed data as median and 95% confidence interval. Receiver operating characteristic (ROC) analyses were used to evaluate the predictive value. To detect statistically significant differences between data sets, a Student–Newman–Keul’s test with one-way analysis of variance was applied to normally distributed variables and a Dunn’s test with one-way analysis of variance on ranks was applied to non-normally distributed variables. A *p*-value < 0.05 was considered statistically significant. Statistical analyses were performed with SigmaStat 3.5 (Systat Software Inc., San Jose, CA, USA) and GraphPad Prism 7.0 (GraphPad Software Inc., La Jolla, CA, USA).

## 3. Results

### 3.1. Patient Characteristics

A total of 106 patients were included in this study. Their mean age was 63 (±12) years at the time of bronchoscopy. The majority (94%) of the cohort underwent bilateral lung transplantation. The primary causes of lung transplantation were pulmonary fibrosis (54%), COPD (21%), and cystic fibrosis (12%). A total of 286 BAL samples were analyzed in this study, and 25.2% of the BAL samples were derived from patients with a CLAD diagnosis. Of these CLAD-BAL specimens, 14.7% came from patients categorized as CLAD stadium 1, 7.7% were categorized as CLAD stadium 2, 1,8% were categorized as CLAD stadium 3, and 1% were categorized as CLAD stadium 4. Only two patients suffered from RAS in the study cohort. Depending on disease severity (CLAD stadium), the maximum forced expiratory volume (FEV1), total lung capacity (TLC), and forced vital capacity (FVC) decreased significantly (Figure 1).

### 3.2. BAL Characteristics

Overall, BAL cell count did not differ between CLAD stages (Figure 2A). However, the percentage of neutrophils present in the BAL fluid was significantly higher in samples obtained from lung transplant recipients with a CLAD stage 1 diagnosis than those without CLAD (Figure 2B). We observed no significant difference in the percentage of macrophages present in the BAL fluid among the different CLAD stages (Figure 2C). Interestingly, the percentage of eosinophils was significantly lower in samples without CLAD than in those with CLAD stage 3/4 (Figure 2D). In progressive CLAD at stage 3/4, the percentage of eosinophils was significantly higher than in CLAD stage 2 (Figure 2D). No significant differences were observed in the percentage of lymphocytes and B-cells among CLAD stages (Figure 2E,F).

Next, we explored the levels of pro- and anti-inflammatory cytokines present in the BAL fluid of the lung transplant recipients. BAL fluid derived from patients with CLAD stage 1 displayed significantly increased concentrations of IL6 over patients without CLAD (Figure 3A). Similar results were obtained with IL8, where transplant recipients without CLAD had significantly lower IL8 levels compared to CLAD stages 1 and 3/4 (Figure 3B). The level of MIP-1⍺ was significantly higher in BAL fluid obtained from patients with CLAD stage 1 than those without CLAD (Figure 3C). 

### 3.3. Mitochondrial Biomarkers in BAL Fluid

Next, we explored the mitochondrial biomarkers ND1-mtDNA and formyl-methionine (f-Met) in the BAL fluid of lung transplant recipients. The level of ND1-mtDNA was significantly lower in patients without CLAD than in those with CLAD (Figure 4A). BAL fluid obtained from patients with CLAD stage 3/4 had significantly higher mtDNA concentrations than those with CLAD stage 1 or without CLAD (Figure 4A). Overall, ND1-mtDNA levels differed significantly between all CLAD stages except for CLAD stages 2 and 3/4. Intriguingly, ND1-mtDNA levels did not differ significantly between patients who presented with acute allograft rejection and those who did not (Figure 4B).

We focused next on levels of f-Met in BAL fluid since it is an important factor for mitochondrial protein translation. We found significantly increased concentrations of f-Met in patients with CLAD stage 3/4 than in patients with CLAD stage 1 or without CLAD (Figure 4C). Furthermore, we observed a significant correlation between levels of ND1-mtDNA in BAL fluid and FEV1 (*r* = −0.3843, *p* < 0.0001; Figure 4D). Moreover, when we compare ND1-mtDNA levels in patients with and without CLAD, the area under the ROC (AUROC) supports a significant increase in patients with CLAD over those without CLAD (AUROC = 0.75, *p* < 0.0001; Figure 4E).

### 3.4. Correlation Matrix Analysis of Mitochondrial and Inflammatory Biomarkers

To obtain further insight into the role of mitochondrial and inflammatory biomarkers in the clinical presentation of CLAD in lung transplant patients, we performed a correlation matrix analysis based on the clinical data (Figure 5). We observed significant positive correlations between CLAD status and ND1-mtDNA, f-Met, neutrophil percentage, and pro-inflammatory cytokine concentrations in BAL fluid. As expected, we also observed a significant negative correlation between CLAD status and FEV1.

## 4. Discussion

CLAD is a life-threatening complication for lung transplant patients [1,2,3]. Without suitable therapeutic options, early diagnosis is critical. In addition to clinically established inflammatory biomarkers, the predictive power of cellular and plasmatic immune system components for CLAD has been assessed. However, they were found to lack sufficient power to be informative [5,12,15,21,22]. Mitochondrial DNA is known to be a damage-associated molecular pattern (DAMP) that is released during acute and chronic inflammatory disease and primary graft dysfunction following lung transplantation. However, it has yet to be evaluated in the context of CLAD [23,24].

Around 25% of the BAL fluid samples analyzed in this study were obtained from patients diagnosed with CLAD, and their distribution of CLAD stages was comparable to other published studies [25]. Moreover, FEV1, FVC, and TLC were observed to decrease as a function of CLAD stage. We investigated several promising surrogate biomarkers for CLAD. Firstly, we were unable to differentiate CLAD stages based on total cell count. However, we found that the percentage of eosinophils and neutrophils increased significantly between non-CLAD and CLAD stage 1 (Figure 2B,D) and between non-CLAD and CLAD stage 3/4 (Figure 2D), consistent with earlier studies [26]. Indeed, a recent study of 36 lung-transplant recipients identified an increase in neutrophils as a potential indicator for CLAD [27]. However, the CLAD and non-CLAD patient data used in both studies overlapped appreciably. Consequently, the evaluation of these significant correlations in a larger, independent data set is mandatory. Secondly, the concentrations of cytokines IL6, IL8, and MIP1⍺ in the BAL fluid increased significantly between non-CLAD and CLAD stage 1 patients (Figure 3A–C). The concentration of IL8 was also significantly elevated in CLAD stage 3/4 patients compared to non-CLAD patients (Figure 3B). Both IL6 and IL8 are known to increase in the early post-transplant period and to be associated with adverse outcomes such as primary graft dysfunction and extracorporeal membrane oxygenation (ECMO) intervention [28,29,30,31,32]. IL6 is an early biomarker for inflammatory complications such as infection or acute rejection and might, therefore, not be a suitable candidate for predicting long-term complications [33]. However, IL8 is a known neutrophilic inflammatory indicator expressed in a donor’s lungs during CLAD and is associated with gastric regurgitation, making it a potential therapeutic target [32,34,35]. MIP1⍺ is another pro-inflammatory cytokine that has yet to be evaluated as a biomarker for CLAD detection. However, since MIP1⍺ levels were only slightly elevated at CLAD stage 1, it would not be suitable as a CLAD biomarker. This observation is consistent with its primary functions in the early inflammatory response and not in later chronic inflammatory states [36,37].

Since BAL cell count and differentiation and the investigated cytokine levels were insufficient to predict CLAD status, this study explored f-Met and ND1-mtDNA levels as potential mitochondrial biomarkers for CLAD. Both are DAMPs associated with neutrophil-mediated lung injury, while their levels during CLAD were unknown [38,39,40]. Although f-Met levels increased significantly between CLAD stages 1 and 4, the variability and overlap of data for each stage made it difficult to differentiate CLAD stages. However, the concentration of ND1-mtDNA increased with each successive CLAD stage, making it a potential CLAD biomarker. Only mtDNA was able to separate CLAD subgroups by severity, whereas the other potential biomarkers (IL6, IL8, MIP1⍺, and f-Met) can separate CLAD vs. non-CLAD but cannot further separate the CLAD subtypes by severity. To substantiate this observation, we performed correlation analyses with FEV1, an important indicator of CLAD, and found a strong inverse correlation between FEV1 and BAL fluid ND1-mtDNA levels. Importantly, BAL fluid ND1-mtDNA levels were moderately predictive of CLAD in the ROC analysis. Lung transplant recipients are endangered by chronic regurgitation yielding progressive lung injury, which might be associated with a general release of mtDNA. Since kidney and lung injury are closely connected and associated with the release of DAMPs, the presence of kidney disease might also bias the predictive power of ND1-mtDNA [41]. It must also be noted that the ROC analysis could only differentiate between the presence and absence of CLAD. Larger sample sizes will be required to explore the predictive power of ND1-mtDNA in discriminating between CLAD stages. To the best of our knowledge, we are the first to investigate ND1-mtDNA levels in the BAL fluid of lung transplant recipients. The only study was that of Yang et al., who demonstrated that the predictive power of cell-free DNA of non-mitochondrial origin in BAL fluid is superior when combined with the cytokine CXCL10 levels (ROCAUC for BOS = 0.8571, RAS = 0.8500, and stable allograft function = 0.8679) [15].

This study did not explore the underlying mechanisms, but its findings are nonetheless explainable. CLAD is an umbrella term for different etiologies that decrease graft function and have heterogeneous pathologies [32]. Many immunologic biomarkers depend on distinct inflammatory origins, explaining their varying concentrations in CLAD patients [12]. However, ND1-mtDNA in BAL fluid might indicate damage to the alveolar cells and, therefore, be distinctive for the lung injury and independent of the underlying pathology of CLAD or other comorbidities. This association might also explain the positive correlation observed between ND1-mtDNA levels and the severity of CLAD. Furthermore, while ND1-mtDNA has been identified as an important activator of acute and chronic inflammation, its role as a DAMP during chronic rejection of transplanted lungs has not been thoroughly investigated [22,40,42]. While the release of mtDNA is known to be triggered by the ischemia and reperfusion injury and to induce a neutrophil-dependent ROS-mediated inflammatory reaction during acute rejection following lung transplantation, analogous mechanisms might contribute to the development of CLAD [22]. Interestingly, ND1-mtDNA BAL levels were not associated with acute rejection of the transplanted lung. In general, the release of mtDNA, neutrophil extracellular traps, and other DAMPs are closely connected to the early inflammatory response after lung transplantation, including acute organ rejection [23]. For example, a recent study showed a positive correlation between primary graft dysfunction and the amount of mtDNA derived from the perfusion fluid of a donor’s lung [24]. Another study revealed an mtDNA-induced activation of neutrophil extracellular traps via the toll-like receptor 9 pathway in transplanted lungs [18]. The opposing findings in our study remain unclear because the underlying measurement techniques did not differ between patients suffering from primary graft dysfunction or CLAD. To the knowledge of the authors, BAL-derived ND1-mtDNA has not yet been specifically evaluated in the context of CLAD, which might indicate that it offers differing distributive patterns during primary graft rejection. For this reason, future studies must clarify how ND1-mtDNA is released into the plasma and alveolar space, not only during acute organ rejection but also during CLAD. The potential link between CLAD and the release of mtDNA might reflect the well-described IL-8-induced neutrophilic shift in the BAL of CLAD, which has also been observed here [26,28,43,44]. Activation of neutrophils can be caused by the release of nucleic acids activating the recognition of pattern-recognition receptors, leading to TLR-dependent mobilization of antigen-presenting cells [45,46]. However, it remains to be clarified whether mtDNA can also induce these pathways.

There are a number of limitations to this study. Firstly, no sample size calculation was performed due to its explorative nature. Secondly, despite sourcing data from two transplantation centers, the number of available patients was limited by the rarity of the underlying diseases. Both limitations must be taken into consideration when interpreting the statistical significance of findings in situations where the interquartile ranges overlap (e.g., neutrophil and eosinophil counts). While our results agree with earlier studies, there remains an urgent need to confirm them in a larger cohort [26,27,35,36]. In addition, this study did not perform a joint analysis of the explored biomarkers. As mentioned above, CLAD consists of multiple diseases that are hard to identify and distinguish with a single biomarker. For this reason, a multimarker approach combining information across markers would be more suitable [12].

Finally, this study demonstrated that mtDNA in BAL fluid has the potential to act as a biomarker for the early detection of CLAD and differentiation of its different stages, independent of the underlying pathology.

## Figures and Tables

**Figure 1 jcm-11-04142-f001:**
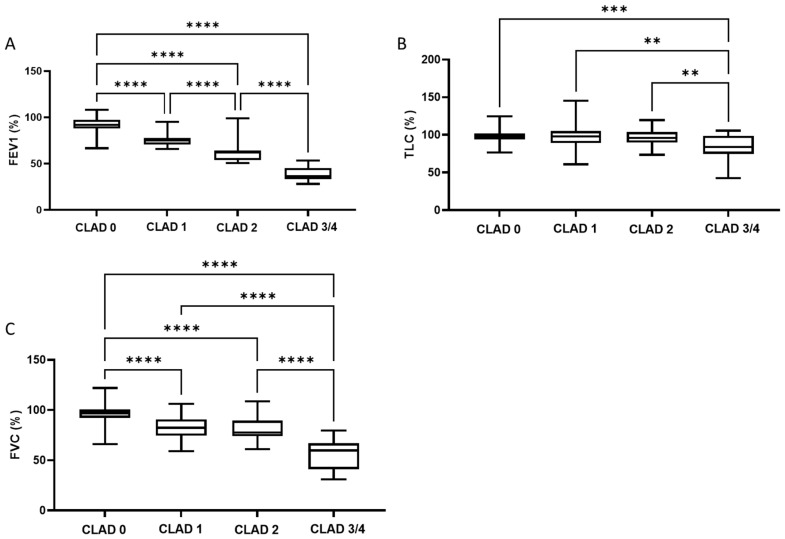
Boxplots showing the relationship between FEV1 (**A**), TLC (**B**), and FVC (**C**), and the CLAD stadium. ** = *p* < 0.01; *** = *p* < 0.001; **** = *p* < 0.0001. Abbreviations: CLAD = Chronic Lung Allograft Dysfunction; FEV1 = Forced Expiratory Volume in 1s; FVC = Forced Vital Capacity; TLC = Total Lung Capacity.

**Figure 2 jcm-11-04142-f002:**
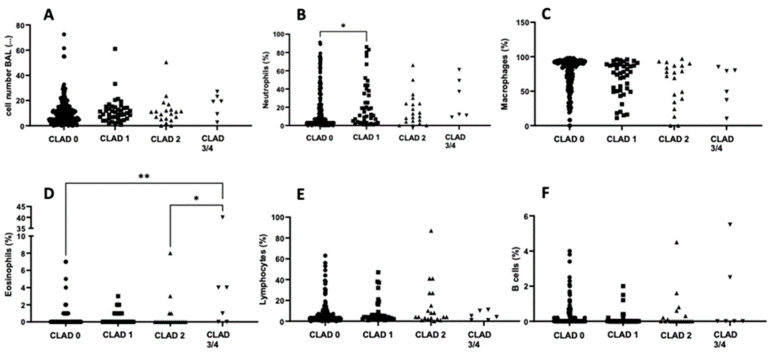
Cellular distribution of BAL fluid among the different CLAD stages. Scatterplots showing the cellular distribution of the analyzed BAL fluid for each CLAD stadium: cell number (**A**), proportion of neutrophils (**B**), macrophages (**C**), eosinophils (**D**), lymphocytes (**E**), and B cells (**F**). * = *p* < 0.05; ** = *p* < 0.01. Abbreviations: BAL = Bronchoalveolar Lavage; CLAD = Chronic Lung Allograft Dysfunction.

**Figure 3 jcm-11-04142-f003:**
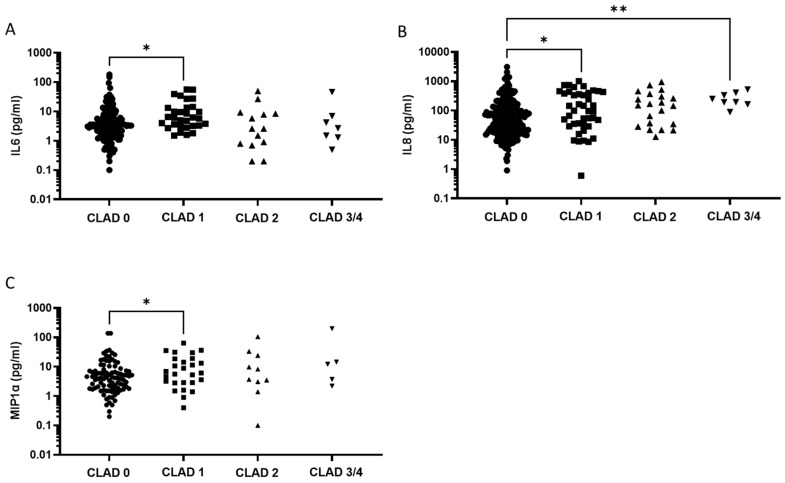
Cytokine concentration in BAL fluid. Scatterplots showing BAL cytokine concentrations for each CLAD stadium: IL6 (**A**), IL8 (**B**), and MIP1⍺ (**C**). * = *p* < 0.05; ** = *p* < 0.01. Abbreviations: CLAD = Chronic Lung Allograft Dysfunction; IL6 = Interleukin 6; IL8 = Interleukin 8; MIP1⍺ = Macrophage Inflammatory Protein-1.

**Figure 4 jcm-11-04142-f004:**
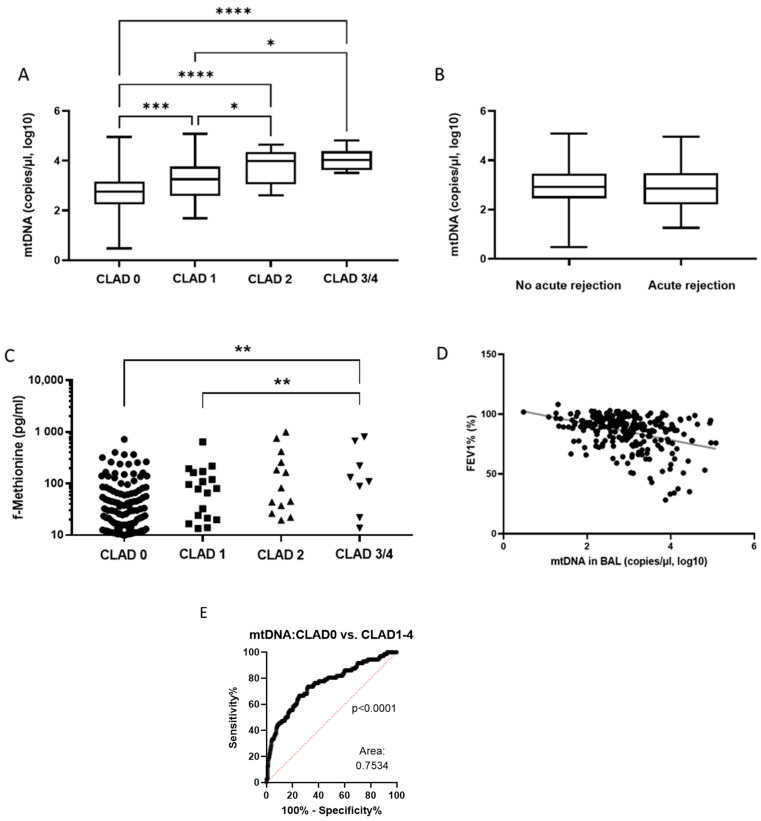
Diagrams showing the relationship between concentrations of ND1-mtDNA (**A**) and f-Met (**C**), and CLAD stage. Boxplot comparing ND1-mtDNA concentration with the status of acute rejection (**B**). Scatterplot showing the correlation between the concentration of ND1-mtDNA and FEV1 (**D**). A ROC analysis of the predictive power of ND1-mtDNA concentration for CLAD status (**E**). * = *p* < 0.05; ** = *p* < 0.01; *** = *p* < 0.001; **** = *p* < 0.0001. Abbreviations: CLAD = Chronic Lung Allograft Dysfunction; FEV_1_ = Forced Expiratory Volume in 1 second; mtDNA = mitochondrial DNA.

**Figure 5 jcm-11-04142-f005:**
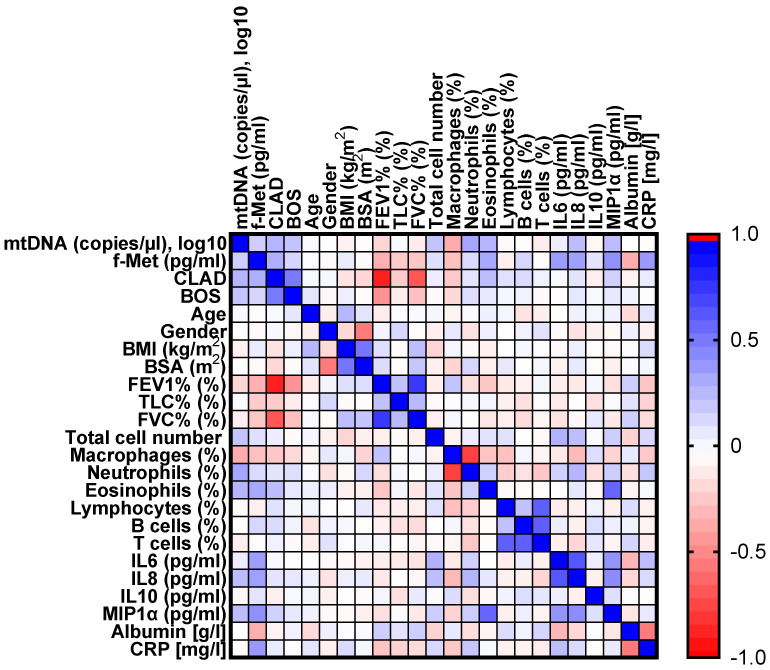
Heatmap of the Pearson correlation matrix. The color depth represents the strength of the correlation between each pair of parameters considered. Blue represents a positive correlation, and red represents a negative correlation. Abbreviations: BMI = Body Mass Index; BOS = Bronchiolitis Obliterans Syndrome; BSA = Body Surface Area; CLAD = Chronic Lung Allograft Dysfunction; C-Reactive protein; FEV1 = Forced Expiratory Volume in 1 second; f-Met = formyl-methionine; FVC = Forced Vital Capacity; IL = Interleukin; MIP1⍺ = Macrophage Inflammatory Protein-1; mtDNA = mitochondrial DNA; TLC = Total Lung Capacity.

## Data Availability

The data presented in this study are available on request from the corresponding author. The data are not publicly available due to privacy and ethical concerns.

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
