# Peer review of "Chronic Lung Allograft Dysfunction Is Associated with Increased Levels of Cell-Free Mitochondrial DNA in Bronchoalveolar Lavage Fluid of Lung Transplant Recipients"

_jcm, 2022, doi:10.3390/jcm11144142_

Round 1
Reviewer 1 Report
Schneck et al submitted a manuscript ‘ Chronic lung allograft dysfunction is associated with increased 2 levels of cell-free mitochondrial DNA in bronchoalveolar lav-3 age fluid of lung transplant recipients.’
In this single center retrospective review they looked a their post lung transplant patients between 2010-2019, and compared the BALF between patients with CLAD vs non-CLAD patients, using previously described potential biomarkers such as various inflammatory cell types (neutrophils, macrophages, eosinophils, B cells), cytokines (IL6, IL 8, MIP-1 alpha).
The novelty of this paper is they then looked at the DAMP between CLAD vs non-CLAD BALF samples, specifically mitochondrial biomarkers, mitochondrial DNA encoding for NADH dehydrogenase 1 and formyl-methionine. The found that the level of ND1-mt DNA correlated not only between CLAD vs non-CLAD BALF samples, but the concentration of ND1-mtDNA increased with increasing severity of CLAD grading. Furthermore the ND1-mtDNA also correlated well with the clinical phenotypes of FEV1, FVC and TLC.
As far as I can tell from my brief literature search the use of mt-DNA as a biomarker for CLAD has not been reported before, and as pointed out by the authors, this is biologically plausible.
Obviously, by the nature of lung transplantation, the number of patients and BALF samples are relatively small, and the results of this study will need to be further validated in a larger patient population to demonstrate convincingly the value of this potential biomarker. Despite these limitations, I believe this manuscript will be of interest to the lung transplant community.
Specific comments:
Line 166
the number of samples from CLAD 3 and 4 were very few in number (1.8% and 1%)
maybe the authors should just consider grouping all the CLAD samples together?
Please see my comments later.
Line 170
pls define how patients with CLAD are graded? Also pls provide the reference for the grading system used for the severity of CLAD used here
Lines 178-194
is there any advantage in subdividing the CLAD patients into the four subgroups? Maybe it is better simply to group all the CLAD cases together and compare with non-CLAD?
if you like, for IL-8 you can then say that there is a difference between CLAD 1 vs CLAD 0, as well as CLAD 3/4 vs CLAD 0.
Figure 4
Following on from my comments above, then for mtDNA, here you can show the more detailed analysis and show the significant difference between CLAD 0 vs each subgroup of CLAD 1-4.
if the intention is to show that mtDNA is a better performing biomarker than different cell types and cytokines in BALF, you can simply make a statement as follows:
Only mt DNA can separate CLAD subgroups by severity, whereas the other potential biomarkers (here you can list them individually) can separate CLAD vs non-CLAD, but cannot further separate the CLAD subtypes by severity. If possible, the detailed graphs showing the lack of discrimination for the CLAD subgroups can be shown as supplementary figures.
Author Response
Answers to the reviewers
The authors would like to thank the reviewers for their excellent suggestions and efforts to increase the quality of our work. The abstract and the manuscript were carefully revised by following the suggested improvement proposals. The authors substantially increased the quality of the manuscript and hope to comply now to the requirements of “Journal of Clinical Medicine”.
Reviewer 1:
Reviewer´s remark:
Line 166: the number of samples from CLAD 3 and 4 were very few in number (1.8% and 1%)
maybe the authors should just consider grouping all the CLAD samples together? Please see my comments later.
Line 170: pls define how patients with CLAD are graded? Also pls provide the reference for the grading system used for the severity of CLAD used here
Lines 178-194: is there any advantage in subdividing the CLAD patients into the four subgroups? Maybe it is better simply to group all the CLAD cases together and compare with non-CLAD?
Figure 4: Following on from my comments above, then for mtDNA, here you can show the more detailed analysis and show the significant difference between CLAD 0 vs each subgroup of CLAD 1-4.
Answer:
Thank you for your vital suggestions. Since the above-mentioned four comments are closely connected, the authors combine their answers in the following section:
As stated in the methods section, patients are scheduled on a regular base for post-transplantation care. Throughout these appointments careful anamnesis on the patient´s exercise capacity, general pulmonary function, and signs of infections is taken. Further, blood analysis, 6-minute-walk, and most importantly spirometry for the staging of CLAD is performed. Based on the spirometry results, CLAD is categorized according to the pulmonary council of the International Society of Heart and Lung Transplantation (ISHLT) into five stages: stage 0: FEV1 >80%; stage 1: FEV1 66–80% baseline; stage 2: FEV1 51–65%; stage 3: FEV1 35–50%, stage 4: FEV1 <35% (referred to baseline FEV1). The according reference (Verleden GM, Glanville AR, Lease ED, et al. Chronic lung allograft dysfunction: Definition, diagnostic criteria, and approaches to treatment-A consensus report from the Pulmonary Council of the ISHLT. J Heart Lung Transplant 2019;38:493-503. May 2019) was included:
“Based on the spirometry results CLAD was categorized according to the pulmonary council of the International Society of Heart and Lung Transplantation (ISHLT) into five stages: stage 0: FEV1 >80%; stage 1: FEV1 66–80% baseline; stage 2: FEV1 51–65%; stage 3: FEV1 35–50%, stage 4: FEV1 <35% (referred to baseline best FEV1).”
The authors agree that summarizing the CLAD stages would simplify the analysis, however also some important findings would vanish. Caused by the low number of patients presenting with CLAD stages 3 and 4, these were already merged. Still, a significant stage-dependent increase is visible regarding the mtDNA and f-methionine levels, which might be of clinical relevance because there is still no validated biomarker for the purpose of CLAD staging available. This study might therefore open a chance for further evaluation of mtDNA in this context. By summarizing the CLAD stages, this aspect would not be visible. For this reason, the authors would suggest keeping the presented CLAD stages and the presentation of mtDNA in figure 4, whereas they must also emphasize that higher number of patients should be included in validation studies.
Reviewer´s remark:
If you like, for IL-8 you can then say that there is a difference between CLAD 1 vs CLAD 0, as well as CLAD 3/4 vs CLAD 0.
Answer:
The authors mentioned this finding in the result section:
“Similar results were obtained with IL8, where transplant recipients without CLAD had significant lower IL8 levels compared to CLAD stages 1 and 3/4 (Figure 3B).”
Reviewer´s remark:
If the intention is to show that mtDNA is a better performing biomarker than different cell types and cytokines in BALF, you can simply make a statement as follows:
Only mtDNA can separate CLAD subgroups by severity, whereas the other potential biomarkers (here you can list them individually) can separate CLAD vs non-CLAD but cannot further separate the CLAD subtypes by severity. If possible, the detailed graphs showing the lack of discrimination for the CLAD subgroups can be shown as supplementary figures.
Answer:
The authors like to thank the reviewer for the suggestion to specify the discussion. The suggested sentence was added to the discussion:
“Only mtDNA was able to separate CLAD subgroups by severity, whereas the other potential biomarkers (IL6, IL8, and MIP1⍺, and f-Met) can separate CLAD vs. non-CLAD but cannot further separate the CLAD subtypes by severity.”
Due to the strict schedule predetermined by JCM, the authors were unfortunately not able to summarize supplementary graphs for all measured variables (as suggested).
Reviewer 2 Report
The work presented by Schneck et al. reports the potential role of cell-free mitochondrial DNA in CLAD patients after lung transplantation.
The work is well written, the methodology appears correct and the results are original and interesting.
Specific minor comments:
The patients included were affected by CLAD. It is not reported how many were affected by BOS and how many by RAS. These two conditions are different from each other and different mechanisms could underlie the two conditions.
In the discussion the authors should try to interpret the reasons why they found no differences in cell-free mitochondrial DNA in patients with ACR.
What expectations do the authors have about the levels of cell-free mitochondrial DNA in serum and the correlation with the levels found in BAL?
Author Response
Answers to the reviewers
The authors would like to thank the reviewers for their excellent suggestions and efforts to increase the quality of our work. The abstract and the manuscript were carefully revised by following the suggested improvement proposals. The authors substantially increased the quality of the manuscript and hope to comply now to the requirements of “Journal of Clinical Medicine”.
Reviewer 2:
Reviewer´s remark:
The work presented by Schneck et al. reports the potential role of cell-free mitochondrial DNA in CLAD patients after lung transplantation.
The work is well written, the methodology appears correct and the results are original and interesting.
Specific minor comments:
The patients included were affected by CLAD. It is not reported how many were affected by BOS and how many by RAS. These two conditions are different from each other and different mechanisms could underlie the two conditions.
Answer:
Thank you for this important aspect. We had only two RAS patients in the study cohort and added the following sentence in the results section:
“Only two patients suffered from RAS in the study cohort.”
Reviewer´s remark:
In the discussion the authors should try to interpret the reasons why they found no differences in cell-free mitochondrial DNA in patients with ACR.
Answer:
The authors agree and include the following sentence to the discussion:
“Interestingly, ND1-mtDNA BAL levels were not associated to acute rejection of the transplanted lung. In general, the release of mtDNA, neutrophil extracellular traps, and other DAMPs are closely connected to the early inflammatory response after lung transplantation including also acute organ rejection [23]. For example, a recent study showed a positive correlation between primary graft dysfunction and the amount mtDNA deriving of the perfusion fluid of donor lung [24]. Another study revealed a mtDNA-induced activation of neutrophil extracellular traps via the toll-like receptor 9-pathway in transplanted lungs [18]. The opposing findings of our study remain unclear because the underlying measurement techniques did not differ between patients suffering from primary graft dysfunction or CLAD. To the knowledge of the authors, BAL-derived ND1-mtDNA has not yet been specifically evaluated in the context of CLAD, which might indicate that it offers differing distributive pattern during primary graft rejection. For this reason, future studies must clarify how ND1-mtDNA is released into the plasma and alveolar space, not only during acute organ rejection but also during CLAD.”
Reviewer´s remark:
What expectations do the authors have about the levels of cell-free mitochondrial DNA in serum and the correlation with the levels found in BAL?
Answer:
This is an interesting question, which must be addressed by prospective studies. Unfortunately, the authors did not have blood samples of the included patients making further measurements not feasible. Since ND1-mtDNA was yet not quantified in the blood of patients suffering from CLAD, their kinetics remain speculative. However, the assumption of increased plasma mtDNA levels might be reasonable indicated by the occurring inflammatory response which was shown in the lungs. For this reason, this important issue is mentioned in the same section as the prior comment (please see answer above).